

# FOCA: a new quality-controlled database of floods and catchment descriptors in Italy

Pierluigi Claps[1], Giulia Evangelista[1], Daniele Ganora[1], Paola Mazzoglio[1], Irene Monforte[1]

[1] Department of Environment, Land and Infrastructure Engineering, Politecnico di Torino, Torino, 10129, Italy

*Correspondence to*: Pierluigi Claps (pierluigi.claps@polito.it)

**Abstract.** We here present FOCA (Italian FlOod and Catchment Atlas), the first systematic collection of data of Italian river catchments for which discharge historical time series are available. Hydrometric information, including annual maximum peak discharge and average daily annual maximum discharge, is complemented by several geomorphological, climatological, extreme rainfall, land cover and soil-related catchment attributes. All hydrologic information derives from the most recently

released datasets of discharge and rainfall measurements. This paper provides a description of all the raw data and the algorithms used to build the basin attributes dataset, to enhance reproducibility and transferability of the analysis. We also describe the approaches adopted to solve problems encountered during the Digital Elevation Model elaboration in areas characterized by complex morphology. Details about the data quality-control procedure developed to detect and correct errors is also reported. One of the main novelties of FOCA with respect to other national-scale datasets is the inclusion of a rich set

of geomorphological attributes and of extreme rainfall features spatially averaged over the watershed. With this first nationwide data collection (available at https://doi.org/10.5281/zenodo.8060737), a wide range of environmental applications, with particular reference to flood studies, can be undertaken on the Italian territory.

## 1 Introduction

The availability of complete, updated and quality-controlled hydro-geomorphological information over nationwide areas is a

key need in a vast range of applications, from hydrological modelling to hydraulic simulations, and rainfall/runoff analyses. Hydro-geomorphological catchment information, called catchments attributes, can also provide a comprehensive description of the landscape and on how the catchment stores and transfers water (Addor et al., 2017).

The importance of river basin catalogs as a support to water resources management was remarkably stressed by the publication of the Water Framework Directive in 2000. River catchments that cross administrative boundaries can suffer the lack of

incomplete and non-uniform information. To tackle this shortcoming, in early 2000 a European initiative undertook the creation of a pan-European river and Catchment Database (Vogt et al., 2007). In recent years, in several European countries considerable efforts have been dedicated to produce similar datasets at a national-scale level. Relevant examples are the datasets built in the framework of the CAMELS initiative, as in the case of Great Britain (Coxon et al., 2020), France (Andréassian et al., 2021; Delaigue et al., 2022), Switzerland (Höge et al., 2023) and Germany (Loritz et al., 2022). Similar

datasets have been released also in other continents, developed within the same framework, as the United States (Addor et al., 2017), Chile (Alvarez-Garreton et al., 2018), Brazil (Chagas et al., 2020) and Australia (Fowler et al., 2021). Most of these datasets are now linked together, in order to build a "Caravan" (Kratzert et al., 2023). Previous notable examples are the Atlases and the datasets developed in Switzerland (https://hydrologicalatlas.ch/), Austria (Fürst et al., 2009) and Canada (Arsenault et al., 2016). In recent years, other large-scale datasets were developed independently on the CAMELS framework, such as those

related to China (Hao et al., 2021) and the one related to the areas of the upper Danube up to the Austrian-Slovakian borders and some nearby catchments (Klinglet et al., 2021). On a global scale, considerable effort to provide hydrological, climatological and land cover information on a global scale is represented by the work of Linke et al. (2019), who created the HydroATLAS database by mapping key hydrological variables at a 500-meter grid resolution. In this latter work, however,



few hydrological and climatic attributes have been considered, and geomorphological catchment descriptors are completely
missing.

In Italy, as of today, only partial-coverage datasets (both in terms of spatial extent and number of variables) are available. We can mention the datasets developed over North-West of Italy (see e.g. Barbero et al., 2012; Gallo et al., 2013), North-East of Italy (Crespi et al., 2021), North-Central Italy (Pavan et al., 2019) or some studies aimed at mapping few variables conducted over entire Italy (ISPRA, 2005a; ISPRA, 2005b; Claps et al., 2008; Crespi et al., 2018; Braca et al., 2021). Among the above
examples, the database realized within the CUBIST (Characterisation of Ungauged Basins by Integrated uSe of hydrological Techniques) project (Claps et al., 2008) was the only attempt to build a multi-variable national data collection, while all the other collections focus only on one specific topic (rainfall, or discharge, or geomorphology). A complete and updated database of hydro-geomorphological variables related to the main catchments of Italy is therefore still missing. One of the reasons behind this gap is the dismantlement of the National Mareographic and Hydrographic Service (Servizio Mareografico e
Idrografico Nazionale, SIMN) that has led to a federated management of the national monitoring network by 21 different administrative agencies. Nonetheless, the high network density available for rain and stream gauges throughout the country can allow to compile nation-wide catalogs, also taking advantage of the considerable number of local studies performed using systematic samples of hydrological measurements.

A recent effort to collect relevant hydrological information with a national coverage is represented by the work of Claps et al.
(2020a, 2020b, 2020c), in which annual maximum peak discharges (hereafter named peak discharges for brevity) and average daily annual maximum discharges (hereafter named daily discharges for brevity) from 1911 to 2016 for 631 Italian catchments are published, together with some basic geomorphological catchment attributes. This collection is mainly addressed to integrate previously untracked flood discharge measurements, whose historical records were not available in the Hydrological Yearbooks but only in special publications. In this paper we aim to substantially enhance the existing catchment and peak
discharge catalogues by complementing the hydrological data with a large set of climatological, geomorphological, soil and land use attributes, many of them computed from recently-released databases. All data sources used comply with the following criteria: (a) nationwide coverage; (b) consistency in data quality (i.e., no regional or local biases); (c) adequacy of the original resolution in relation to the type of information. All the data sources used in this work can be downloaded from public repositories, except for a few variables that are not easily accessible. In these cases, the original data sources were included in
the dataset created in this work, together with the mean catchment descriptors, to allow replicability. The result of this work is FOCA (Italian FlOod and Catchment Atlas), the first national-scale catchment attributes dataset in Italy (Claps et al., 2023).

The paper is structured as follows: in Section 2 the database history and the rationale used to select catchments to be included in the database are described; in Section 3 and 4 we present the different categories of raw data, that are partly already quality-controlled, which only required some adjustments and substantially concern digital elevation models and streamflow measures,
and partly processed data, deriving from the aggregation and manipulation of rainfall and soil information. In Sections 3 and 4 we also provide an overview of all the geo-morpho-climatic attributes computed for each catchment, together with the methodologies used for their definition and the algorithms used for their evaluation, to grant replicability and thus allowing researchers to perform the same study or extend this one on other catchments. In Section 5 the main characteristics of the FOCA dataset are summarized, and some concluding remarks are presented.

## 2 Presentation of historical catalogs and rationale for catchment selection

Since early '900, the Italian monitoring network was managed by the SIMN. This institution was in charge of data collection, validation, and publication in a series of Hydrological Yearbooks with yearly updates. For each year, two standardized documents (Part I and Part II) were usually published. The first volume (Part I) was dedicated to temperature and precipitation; the second volume contained information and elaborations related to daily and monthly average discharges.



To retrieve an entire time series measured by a gauging station, the consultation of all the Yearbooks was necessary, due to this separation. All the mentioned Yearbooks are available as images (in gif format) on http://www.bio.isprambiente.it/annalipdf/. To facilitate the consultation of discharge data, this information was then processed by the SIMN at monthly time resolution and grouped by individual gauging station, with the aim of providing several years of data in one data sheet for each station. This summarized information, available up to 1970 and complemented with some basic

catchment information, became the "Pubblicazione n°17" (Publication n°17); it also included the annual peak values that were not directly available in the Yearbooks. This document was updated about every 10 years from 1934, and several issues are still available, by progressively including the data acquired up to 1970 (Servizio Idrografico, 1980).

Unfortunately, the update of this publication was interrupted, as about thirty years ago the SIMN was dismantled and 21 different local agencies became responsible of the management of the hydrographic services. Interruption of the publication

of the "Pubblicazione n°17" was particularly negative for the peak discharge data. They have indeed been published only in the "Pubblicazione n°17", as the maxima of the instantaneous discharge and have never been available in the Hydrological Yearbooks. This means that peak discharges measured after 1970 were only available, for many years, only in the regional hydrographic offices. The main effort to recover these unpublished data was done by the GNDCI (Gruppo Nazionale per la Difesa dalle Catastrofi Idrogeologiche) in the VAPI project (http://www.idrologia.polito.it/gndci/Vapi.htm) subsequently

integrated by the CUBIST project.

Within the CUBIST project (Claps et al., 2008) a first, major, advancement in the creation of a national-wide database was achieved. More specifically, historical information available only in printed format were digitized and merged with more recent ones. Moreover, each catchment closed by the gauging stations included in such collection was characterized from a climatological and geomorphological point of view by means of the computation of key catchment attributes.

In the following years, some regional-scale works were conducted to update this systematic collection but without a nation-wide coordination. For example, in North-West Italy, a first follow-up work was the "Catalogo delle portate massime annuali al colmo del bacino occidentale del Po" (Barbero et al., 2012) that includes a considerable number of additional observations from the gauging stations operated by the regional environmental agency as well as additional gauging station managed by other public/private bodies, such as ENEL (a major Italian energy provider) and CNR (Italian Research Council). In this

publication, a systematic and extensive work of data validation was performed to convert into discharge values the hydrometric stage values already recorded by the SIMN but never processed with adequate rating curves. This collection summarizes the information of 140 catchments with at least 5 years of data of peak or daily discharges and contains some basic information of the catchments. Unfortunately, this publication does not contain all the attributes evaluated within the CUBIST project. This regional collection was thus improved with the release of the "Atlante dei bacini imbriferi piemontesi" (Gallo et al., 2013),

that contains geomorphological attributes of about 200 gauged river catchments located over the same region. In this case, however, the peak discharges were not reported as not available for all the basins. Similar works were also carried out over other limited areas, as required by local studied on flood frequency analysis (e.g., Rossi and Caporali, 2010).

Such collections of flood datasets were recently homogenized, integrated and updated by including the most recent data acquired all over Italy, releasing a comprehensive catalogue of floods ("Catalogo delle Piene dei Corsi d'acqua Italiani"; Claps

et al., 2020a, 2020b, 2020c). However, in this case, only basic geomorphological information was included for the considered basins.

The catchment selection criterion adopted in the present work stems from the purpose to improve the work by Claps et al. (2020a, 2020b, 2020c). The 631 chosen catchments are those for which peak or daily discharges are available, and therefore all included in Claps et al. (2020a, 2020b, 2020c). Details on the different sources of the historical discharge time series and

their integration are given in Section 4.4. The database features are presented in two different sections. Section 3 discusses the determination and validation of the attributes depending only on the landforms. Section 4 presents various other features mainly obtained through spatial averaging of rasterized information.



## 3 Data and catchment geomorphological attributes

### 3.1 The Digital Elevation Model

As of today, the only national-scale high-resolution Digital Elevation Model (DEM) available for the whole of Italy is the TINITALY/01 (Tarquini et al., 2007). Despite its very high spatial resolution (10 m) that makes its use quite interesting, especially for the delineation of small catchments, it presents the drawback of being obtained by merging separate DEMs of single administrative regions, thus not allowing to work with the same accuracy level nationwide. To overcome this drawback, we adopted the DEM from the Shuttle Radar Topography Mission (SRTM) at 30 meters spatial resolution (Farr et al., 2007).

Processing of DEM and, in general, all the other catchment attributes, has been performed with open-source software, namely GRASS GIS and R. In this work, the original SRTM DEM has been re-projected into the WGS84/UTM zone 32N coordinate system by means of a bicubic interpolation and resampled to obtain integer cells, i.e., the data format requested by the *r.basin* GRASS GIS add-on (Di Leo et al., 2013) used to derive most of the geomorphological attributes, as will be discussed in the following sub-sections. DEMs usually contains pits (i.e, elevation values way lower than those of nearby pixels, that are errors

due to the resolution of the data) that should be filled to ensure a proper delineation of catchment boundaries and drainage networks. Thus, the pit filling procedure was carried out using the *r.hydrodem* GRASS GIS add-on (Lindsay et al., 2005).

   The *r.basin* add-on algorithm requires a series of information to be executed. The input parameters required by *r.basin* for the attribute extraction routine are presented in the following sub-sections.

### 3.2 Catchment boundaries

To determine the basin boundaries, it is first necessary to generate the drainage network from the depitted DEM. This step can be done only once with the r.basin command, which adopts the following steps: (i) calculation of drainage directions with the Multiple Flow Direction (MFD) algorithm; (ii) calculation of the flow accumulation, i.e. the Total Contributing Area (TCA) map; (iii) estimation of stream network, after specifying a threshold value that defines the minimum drainage area required to initiate a channel. In this work threshold values equal to 0.02, 0.1 and 1 $km^2$ are used to extract the stream network for basin

areas smaller than 1 $km^2$, between 1 and 10 $km^2$ and larger than 10 $km^2$, respectively. These values have been identified considering recommendations derived from several research works that investigated the spatial resolution sensitivity of catchment geomorphologic properties and the effect on hydrological simulation, such as Montgomery and Foufoula-Georgiou (1993), Yang et al. (2001) and Beighley and Gummadi (2011). These works highlighted that the river networks generated with larger threshold areas tend to lose relevant information and led us not to use a fixed threshold value, that could be problematic

in small basins.

   To proceed toward delineation, a unified determination of the basin outlet coordinates is necessary, as the official coordinates, taken from hydrometric stations, do not necessarily coincide with stream locations automatically determined from the DEM. So, the coexistence of two sets of coordinates, the real and the operational DEM-based ones, must be properly accounted for in the creation and management of the dataset. The second set of coordinates, the DEM-based ones, are evaluated on the basis

of the reprojected SRTM DEM: in case a new DEM will be used in future works, this set of points needs to be evaluated again. Even though the *r.basin* algorithm is able to automatically work with input outlet coordinates not exactly overlapping with the DEM-based river network, through snapping to the closest point belonging to the network, in some cases the relocation may fail and a manual repositioning of the outlets is thus required. Compared to the real coordinates, the required adjustments are in some cases of the order of a few kilometers. This adjustment is needed to obtain a river network that matches with the

reference one provided by the Istituto Superiore per la Ricerca e la Protezione Ambientale (ISPRA, available on http://www.sinanet.isprambiente.it/it/sia-ispra/download-mais/reticolo-idrografico/view).

   As expected, the procedure for the catchment boundary delineation is generally very accurate where the differences in elevation are quite marked (i.e., the alpine areas). Thus, the only manipulation required is the repositioning of the catchment outlet, in

Earth System
Open Access Science
Data Discussions

our case always positioned on the gauging stations. However, in flat areas this method does not always provide the real drainage

directions and it is therefore necessary to manually force the DEM to correct the Total Contributing Area (TCA) map built by the *r.basin* procedures, a practice commonly known as *stream-burning* (Lindsay, 2016). This operation is performed individually, when needed, by comparing the unconstrained river network produced by *r.basin* with the reference one provided by ISPRA. This quality checking has been iteratively repeated by carving rivers or inserting artificial barriers to force the stream to the correct path. The extent of this operation i.e., location and length of the carved and/or the walled portions, are

unique for each area: for this reason, this operation requires an individual assessment. The procedure described above is depicted in the flow chart in Figure 1.

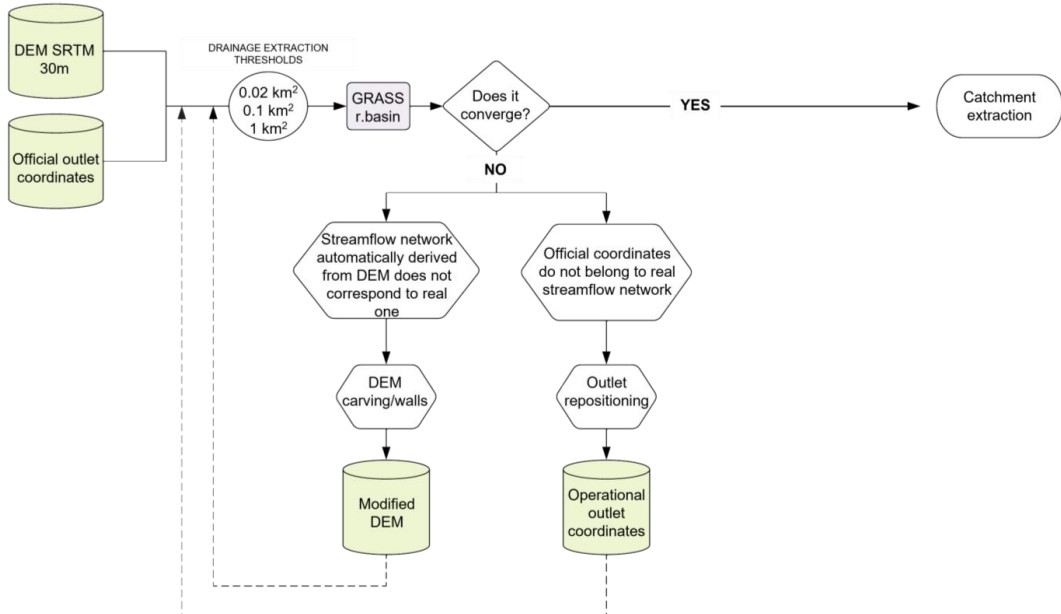

**Figure 1. Visual representation of the workflow adopted to delineate catchment boundaries.**


The catchment boundaries resulting from the delimitation are finally made available in vector format in the WGS84 UTM32 N (EPSG 32632) coordinate system. The catchment boundaries were used as masks to clip several layers of climatological and soil-related attributes. Moreover, the provision of this geographic information will allow users to expand this database by computing other descriptors using possible new gridded datasets.

**3.3 Geomorphological catchment attributes**

Geomorphoclimatic information, combined with soil characteristics, is essential to try to characterize how the catchment stores and transports water, both on and below the surface. In this work, we have first selected from the literature a large set of morphological attributes that can be directly obtained by processing the 30-meter resolution STRM DEM. Automatic procedures adapted around the GRASS add-on *r.basin* is used for this computation, as *r.stat*, *r.slope.aspect*, *r.stream.stats*

(Jasiewicz, 2021) and *r.accumulate* (Cho, 2020) functions, obtaining 61 geomorphological attributes (see Table 1 for a complete description).

| Attribute sub-category | Attribute | Notation | Units | Description |
|---|---|---|---|---|
| Altimetric and geometrical | Area | area | km² | Catchment area computed by multiplying the area of a single pixel by the number of pixels within the catchment boundary. |
| | Mean Elevation | elev_mean | m a.s.l. | Catchment mean elevation. |
| | Maximum Elevation | elev_max | m a.s.l. | Catchment maximum elevation. |
| | Minimum Elevation | elev_min | m a.s.l. | Catchment minimum elevation. |
| | Aspect | aspect | ° | Mean of the angle of exposure on the horizontal plane of each cell of the catchment. The adopted convention is that North is 0° and the aspect is computed clockwise. |
| | Hypsographic curve | elev_x | m a.s.l. | Elevation values of the hypsographic curve (i.e., the curve that defines the distribution of catchment areas located within a specific elevation range). Each $x$ corresponds to a different percentage of area (2.5, 5, 10, 25, 50, 75, 90, 95 and 97.5%). |
| | Geographic centroid | x_g y_g | m | Coordinates of the pixel nearest to the centroid of the geometric figure resulting from the projection of the catchment on the horizontal plane. |
| | Length of the orientation vector | dir_length | km | Length of the segment linking the catchment centroid to the outlet. |
| | Orientation | orient | ° | Angle of the orientation vector with respect to North. |
| | Mean slope 1 | slope1 | % | Mean slope value calculated averaging the slope map. |
| | Mean slope 2 | slope2 | % | Angle at the base of the right-angled triangle whose base is the square root of the catchment area and twice the median elevation of the catchment (relative to the closing section) as height. This slope is calculated with respect to a catchment of square shape equivalent to the real one and does not consider its actual shape, which can be elongated. |
| Horton Ratios | Horton-Strahler numbers | HS_num_u HS_length_u HS_area_u HS_slope_u | - | 4 sets of $u = 3$ vectors (each corresponding to a Horton order), containing respectively: the number of streams of a given order [-], the average length of the streams of a given order [km], the average contributing area for each order [km²] and the average slope of the streams of each order [%]. Slopes are calculated as ratio of difference of elevation between ends of the segment to its length. |
| | Area ratio | R_a | - | Ratio of the average area drained by streams of a given order $u+1$ and streams of order $u$. |
| | Bifurcation ratio | R_b | - | Ratio of number of stream branches of a given order $u$ to the number of streams branches of the next order $u+1$. |
| | Length ratio | R_l | - | Ratio of average length of streams of two adjacent orders $u$ and $u+1$. |
| | Slope ratio | R_s | - | Ratio of average slope of streams of two adjacent orders $u$ and $u+1$. |
| Streamflow network | Total stream length | TSL | km | Total length of the river network, obtained by summing the length of all its segments. |
| | Drainage density | drain_dens | km⁻¹ | Ratio between the total stream length and the catchment area. |
| | Length of main channel | LMC | km | Length of the longest succession of segments that connect a source to the outlet of the catchment. |
| | Length of longest drainage path | LLDP | km | Path included between the outlet and the furthest point from it, placed on the catchment boundary and identified by following the drainage directions. For most of its length the longest drainage path overlaps the main channel. |
| | Topological diameter | topo_d | - | Number of confluences found on the main channel. |
| | Mean hillslope length | MHL | km | Average of the distances (measured following the drainage directions) of all the pixels not belonging to the hydrographic network, starting from the first pixel of the hydrographic network into which they drain. |
| | Mean slope of longest drainage path | LLDP_slope | % | Calculated as $$LLDP\_slope = \frac{1}{topo\_d}\sum\frac{\Delta z_i}{L_i}\cdot 100 \qquad (1)$$ where topo_d is the topological diameter, $L_i$ is the length of the i-th segment into which $LLDP$ is divided and $\Delta z_i$ is the corresponding elevation difference |
| Shape factor and amplitude function | Shape Factor | shape_f | - | Ratio of catchment area to the square of the length of the main channel. |
| | Elongation Ratio | elong_r | - | Ratio of the diameter of a circle of the same area as the catchment to the maximum drainage path length. |
| | Circularity Ratio | circ_r | - | Ratio between the catchment area to the area of the circle having the same circumference as the perimeter of the catchment. |
| | Compactness coefficient | c_c | - | Ratio of catchment perimeter to the diameter of the circle having the same area of the catchment. |
| | Width function characteristics | width_mean width_var width_skw width_kur width_x | - | Frequency distribution of the distances of each cell of the catchment, along the drainage path, to the outlet. The first four statistical moments (mean, variance, skewness and kurtosis) of this function were calculated as well as the percentiles vector containing the distance to the outlet that includes pixel percentages of x = 5%, 10%, 15%, 30%, 40%, 50%, 60%, 70%, 85% and 95%. |

**Table 1. List of geomorphological attributes, with a brief description and an indication of the algorithm/add-on used for their computation. All the attributes are computed by processing the SRTM DEM at 30 m resolution with the *r.basin* add-on, that takes advantage of other the GRASS GIS algorithms mentioned at the beginning of Section 3.3.**




The provision of this set of 61 different attributes for more than 600 catchments nationwide represents one of the strengths of the FOCA dataset. Comparing to other relevant examples that we found in the literature, such as the CAMELS or the LamaH-CE datasets, one realizes that they include only a limited number of geomorphological descriptors, focusing mainly on climatic, hydrologic, land cover, soil and geological indices. As an example, in the CAMELS developed for the U.S. (Addor et al., 2017) or for Brazil (Chagas et al., 2020), only basic topographic characteristics related to outlet coordinates, catchment area, mean elevation and mean catchment slopes are reported. In the LamaH-CE dataset (Klingler et al., 2021) some additional geomorphological attributes were included, such as the median basin elevation, the range of catchment elevation, the elongation ratio, the horizontal distance from the farthest point of the catchment to the corresponding gauge and the drainage density. The inclusion of other descriptors in LamaH-CE was motivated by the need of knowing the shape of the catchments and the stream network influence on runoff formation. We believe that such a small number of descriptors included in previous works can be insufficient to fully characterize areas with complex river networks. Here we stress the need to carefully characterize the catchments also from a geomorphological point of view, especially when working with a high percentage of catchments located in mountainous areas.

To further highlight the importance of this latter point, we can underline that in the Italian context, that is the focus of this work, non-standard geomorphological parameters have already demonstrated to be significant in several applications of statistical hydrology, as for example the regionalization of the flood frequency curves developed by Laio et al. (2011) over NW Italy. In that case, the proposed regression models consider as covariates geomorphological parameters like the area, the mean elevation, the length of the longest drainage path, the length of the orientation vector, and the catchment outlet coordinates. In a revised version of the methodology (Ganora et al., 2014), also the minimum elevation and the shape factor resulted as significant covariates. In Ganora et al. (2023) a non dimensional flood reduction function to estimate design hydrographs in ungauged catchments can be obtained with a multiple linear regression model that includes the longest drainage path length and slope of the catchment, the average catchment elevation and the fourth statistical moment of the width function.

Moving to a broader context, the literature suggests that among the factors that govern catchment hydrological response in the process of rainfall-runoff transformation, the catchment geomorphological features have been well represented (see e.g. Nagy et al., 2021; Ravazzani et al., 2019). The synthesis of basin response function from physical basin characteristics becomes crucial in ungauged catchments (Singh et al., 2013). Finally, most of the formulations available in the literature to estimate response time when no runoff data is available typically contain a characteristic length and the slope of the catchment or of the main channel (e.g. Chow, 1962; Kirpich, 1940; Sheridan, 1994), which are not easily available in the other above-mentioned databases.

The catchments included in the FOCA dataset cover a wide variety of morphological features and a considerable elevation range (Figure 2a). Figure 2 shows also the empirical cumulative distribution function (ECDF) of some key catchment attributes, such as the area (Figure 2b) and the mean catchment elevation (Figure 2c).

Earth System
Science
Data

Open Access | Discussions

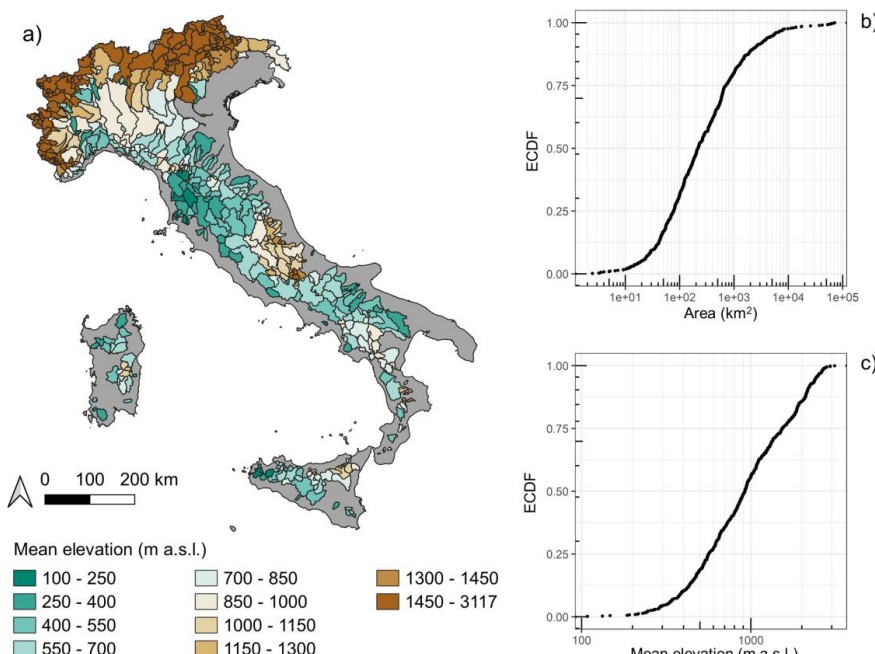

**Figure 2. Mean elevation of the 631 catchment (a) and empirical cumulative distribution function (ECDF) of some key catchment attributes: area (b), mean elevation (c).**

**3.4 Quality check of geomorphic data**

To provide a robust set of catchment features we have checked the consistency between this new dataset and the previously published one (Claps et al., 2020a, 2020b, 2020c) in terms of catchment areas, which were mostly based on the values published in the Hydrologic Yearbooks and in the "Pubblicazione n°17". In order to set the level of accuracy of the procedures that we used, a 10% maximum deviation of the difference (positive or negative) was considered acceptable. All the catchments for

which the discrepancy is greater than 10% have been individually re-examined, as described in Section 3.2. In some cases, the discrepancy was found to derive by a small shift of the catchment outlets upstream or downstream a confluence, thus including or excluding relevant sub-catchments.

Another important check has been made analyzing the difference between the length of main channel (LMC) of each catchment and the length of the longest drainage path (LLDP). Even though a strong relationship between these two attributes is well

known, this connection was not always found in our data: for some catchments the two lengths were significantly different, even after a thorough manual check. A 3-km different threshold (corresponding to the value that select the upper 5% of the catchments with marked differences) was used to identify those catchments for which the difference in length between LMC and LLDP should be further investigated. For these catchments, manual inspections highlighted a drawback in the GIS procedure that produced LMC and LLDP measures with unrealistic discrepancies. Two different cases were observed. In one

case, the resulting main channel shapefile consisted of a polygonal chain made up by multiple features that needed to be merged in one. In the other case, multiple LLDP that differed from each other no more than one hundred meters (2-3 pixels) were identified for the same catchment. In this last case, one LLDP is manually chosen, and the other ones were removed. We also observed that the two situations could also occur simultaneously.

To reach a general feeling on the consistency of these two attributes some scaling laws of the drainage network can be

employed. One of the best-known is the Hack's law equation (Hack, 1957; Eq. (2)), which reads:

$$LLDP = \alpha \, area^{\beta} \tag{2}$$



where the coefficients α and β vary depending on the study area. A log-log scatter plot between the main channel lengths and the basin areas for all the 631 catchments is displayed in Figure 3, where panels (a) and (b) refer to the results before and after the quality control, respectively. By means of the above-mentioned comparisons, we were able to double-check consistency

between the two attributes, recognizing errors up to the order of about 20 km that we checked and corrected.

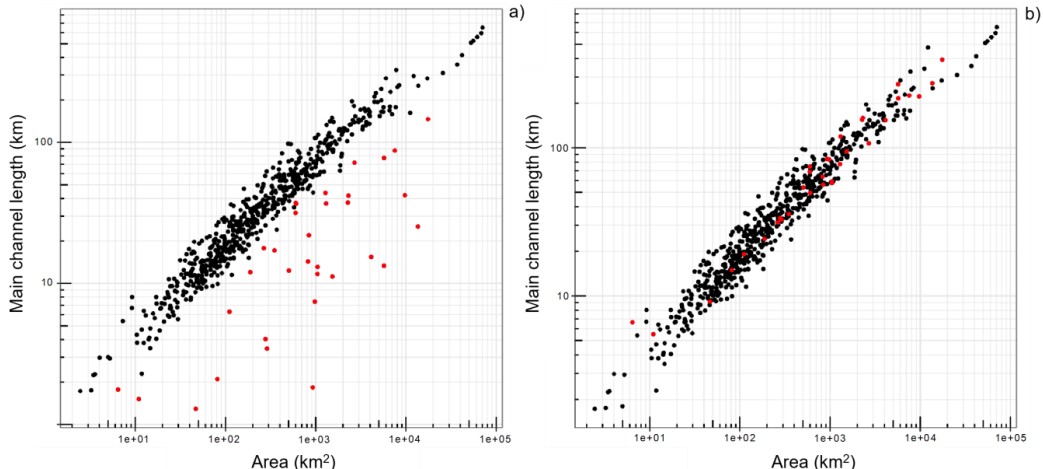

**Figure 3. Scaling law between main channel lengths and basin areas for the 631 catchments before (a) and after (b) the quality-control procedure on LLDP and LMC.**

**4. Data and catchment attributes concerning soil, vegetation and climate**

**4.1 Soil, land cover and NDVI catchment attributes**

In this sub-section, different rasterized information is analyzed to provide area-averaged values. The spatial resolution of the raw data range from 100 to 1000 m and data are not resampled at a unique resolution before computing the average values.

Soil descriptors included in the FOCA dataset can provide information connected to an area-averaged estimate of the soil

permeability conditions. They are the mean areal values of the Curve Number (Soil Conservation Service, 1972) and of the saturated hydraulic conductivity.

The Curve Number is an empirical parameter used to evaluate the portion of total rainfall that becomes net rainfall in a flood event. The values used in this study were taken from a national scale cartography produced by Carriero (2004) at 250 m resolution, consistently with the work of Ganora et al. (2013). Since this dataset is currently not available online, it was included

in the dataset also in the form of a raster map to grant replicability and thus allowing researchers to perform the same study or extend this one on other catchments. Three types of Curve Numbers are defined, which vary according to the antecedent wetness conditions (i.e. dry, average or wet) of the soils. For each Curve Number type, we provided the mean value and the (spatial) coefficient of variation. It is worth highlighting that all the coefficients of variation reported in this work are referred to the spatial variability of an attribute within the catchment, while its temporal variability is not considered here. The only

exception is the coefficient of variation of rainfall regimes, that will be introduced later, whose description is reported in Table 4.

To derive a feature that approximates soil permeability features we started from soil texture fractions maps derived from SoilGrids (Hengl et al., 2017; available on: https://soilgrids.org/). These cartographies map the spatial distribution of soil properties across the globe at 250 m spatial resolution, at seven standard depths, from 0 cm to 200 cm. SoilGrids maps are

based on over 230,000 soil profile observations from the WoSIS (World Soil Information Service) Database (Batjes et al., 2009). Soil texture information were derived from these maps averaged over the first 30 cm of depth, a value consistent with





the hydrological purposes of this work. Based on sand and clay content, we derived the saturated hydraulic conductivity using the pedo-transfer function proposed by Saxton et al. (1986; Eq. (3)).

The 7 soil descriptors (6 related to the Curve Number and one to the saturated hydraulic conductivity) that we have considered

are listed in Table 2.

| Soil attributes | | | | |
|---|---|---|---|---|
| Attribute sub-category | Attribute | Notation | Units | Description |
| Soil | Curve Number | CN1, CN1_cv CN2, CN2_cv, CN3, CN3_cv | - | Empirical parameter developed by the Soil Conservation Service (1972) and used to predict direct runoff, whose value is between 0 and 100. According to the antecedent moisture condition that refers to the preceding wetness condition of soils, CN is divided into three classes, namely CN1, CN2, and CN3. CN2 is the average condition, while CN1 and CN3 represent the lowest (dry soil) and highest (saturated soil) runoff potentials, respectively. |
| | Saturated Hydraulic Conductivity | k | cm/d | Computed from sand and clay content of SoilGrids maps at 250 m resolution as follow: $$k_s = 24e^{\left[12.012 - 7.55 \cdot 10^{-2}\, s + \frac{(-3.895 + 3.671 \cdot 10^{-2}\, s - 0.1103 c + 8.7546 \cdot 10^{-4}\, c^2)}{0.332 - 7.251 \cdot 10^{-4}\, s + 0.1276 \log(c)}\right]} \qquad (3)$$ where s is the sand content (%) and c is the clay content (%). |

**Table 2. List of soil attributes.**

Figure 4 provides a snapshot map of the catchment-averaged soil parameter values. The spatial distribution resulting for the

saturated hydraulic conductivity (Figure 4a) illustrates significantly different conditions in the soil properties moving from North to South of Italy and reflects the high clay content characterizing the soils of the Apennine basins. The same level of difference is not visible in the CN2 values (Figure 4b), essentially because they reflect the geologic and land-use (not soil) information.

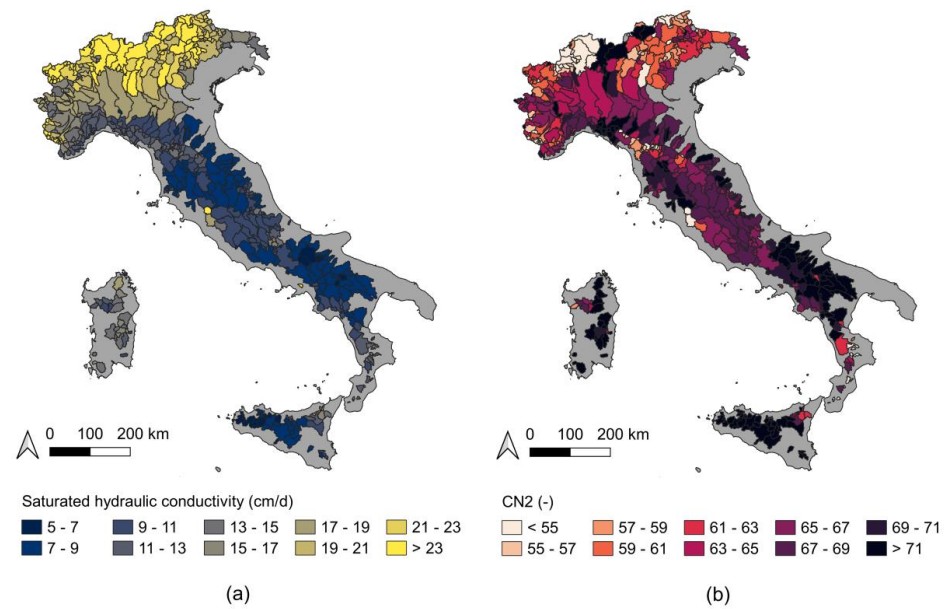


(a)                    (b)

**Figure 4. Mean areal values of the saturated hydraulic conductivity (a) and the Curve Number in average conditions, i.e. CN2 (b). Administrative boundaries: GADM v3.6.**

Land cover characteristics at 100 m resolution are extracted from the 44 classes of the third level of CORINE Land Cover

2018 (available on: https://land.copernicus.eu/). In particular, 5 land indices are considered here, as obtained by merging



similar classes. The 5 land indices are outlined in Table 3. A map of one of the classes is provided in Figure 5a: it displays an overview of the percentage of urbanized areas, thus providing some information related to the anthropogenic settlement relevance within the catchments. As expected, the vegetation coverage is lower in high elevation and cold regions, i.e. the Alps, that are also regions with low percentages of urbanized areas.

We also computed multi-temporal indicators of the NDVI (Normalized Difference Vegetation Index) maps, whose data are provided by the Copernicus Land Monitoring Service (available on https://land.copernicus.eu/). NDVI is an index that shows if the area under investigation contains live green vegetation and its overall health. We used the values of the Long Term Statistics (LTS) NDVI V3.0.1 of the Copernicus service, with a 1 km spatial resolution, that are NDVI mean observations over the period 1999-2019 for each of the 36 ten-daily periods of the year, resulting in 36 raster maps. These maps were used to

compute the mean annual NDVI value, the (spatial) coefficient of variation of the mean annual NDVI and the spatio-temporal mean NDVI regime. The NDVI regime is intended as the the diagram representing the multitemporal mean on 36 time intervals of 10 days. To synthetically characterize the latter, a Fourier series representation was used, which allows the shape of the regime to be described with fewer parameters (4, in total) than the 36 10-day average values of which it is made up. A more detailed description of the four parameters that describe the coefficients of the Fourier series that represent the NDVI regimes

is reported in Appendix A.

After this data preparation, catchment boundaries were used to extract a total of 11 land cover and NDVI attributes, listed in Table 3. The mean NDVI is mapped in Figure 5b to provide some insights of the variation of the mean greenness of the biomes.

## Land cover and NDVI attributes

| Attribute sub-category | Attribute | Notation | Units | Description |
|---|---|---|---|---|
| Land cover | Corine Land Cover 1 | clc1 | % | Percentage, on the catchment area, of continuous and discontinuous urbanized areas (CORINE classes 111, 112). |
| | Corine Land Cover 2 | clc2 | % | Percentage, on the catchment area, of woods (311, 312, 313), arboreal vegetation, shrub vegetation, bushes (CORINE classes 324, 323, 321, 322). |
| | Corine Land Cover 3 | clc3 | % | Percentage, on the catchment area, of herbaceous vegetation, meadow pasture, special crops, olive groves, vineyards, arable land (CORINE classes 231, 222, 223, 221, 211,241, 243, 242, 142). |
| | Corine Land Cover 4 | clc4 | % | Percentage, on the catchment area, of non-vegetated areas (331, 333, 332, 334), mining areas, landfills, and construction sites (CORINE classes 131, 133), industrial and commercial areas, communication networks (CORINE classes 121, 122, 123, 124). |
| | Corine Land Cover 5 | clc5 | % | Percentage, on the catchment area, of humid areas (CORINE classes 411, 512, 521). |
| NDVI | NDVI | NDVI NDVI_cv | - | Indicator of the greenness of the biomes measured by satellite, whose value is between 0 and 1. It is defined as $$NDVI = \frac{REF_{nir} - REF_{red}}{REF_{nir} + REF_{red}} \qquad (4)$$ where $REF_{nir}$ and $REF_{red}$ are the spectral reflectances measured in the near infrared and red wavebands respectively. Mean value and (spatial) coefficient of variation were computed. |
| | B1, B2, C1, C2 | B1_NDVI, B2_NDVI, C1_NDVI, C2_NDVI | - | Mean values of the coefficients of the Fourier series representation of NDVI (see Appendix A). |

Table 3. List of land use and NDVI attributes.




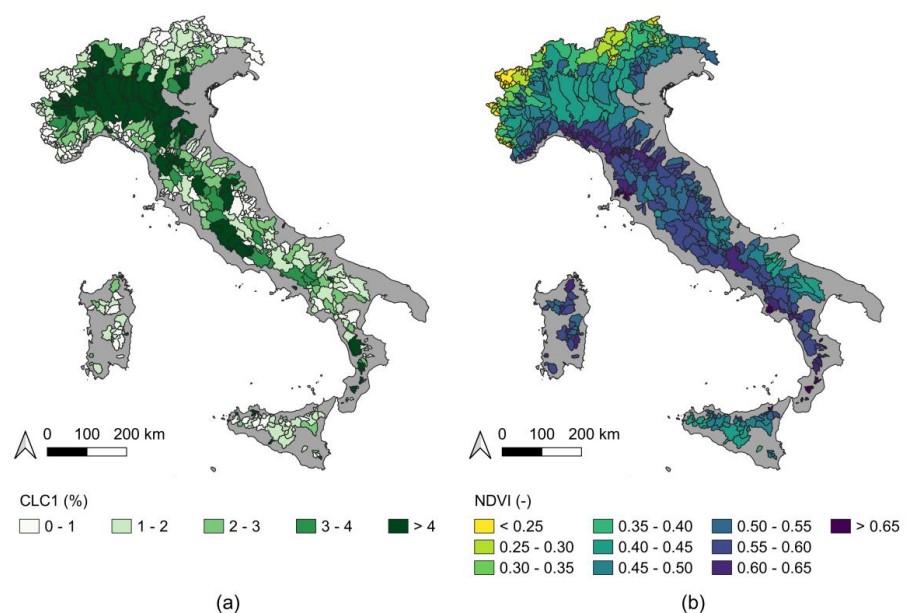

(a)                                                                    (b)

**Figure 5. Spatial variability of the mean areal NDVI (a) and percentage of clc1 (b). Administrative boundaries: GADM v3.6.**

### 4.2 Climatological catchment attributes

State-of-the-art national-scale datasets at 1 km resolution have been used for the evaluation of several climatological attributes,
described below.

Mean monthly precipitation information is taken from the BIGBANG (Bilancio Idrologico GIS BAsed a scala Nazionale su
Griglia regolare; Braca et al., 2021) 4.0 dataset, which covers the 1951-2019 period and is obtained by means of spatial
interpolation at 1 km resolution of rain gauge measurements, integrating, only over limited areas and for certain years, the
spatial interpolation produced by ARCIS (Archivio Climatologico per l'Italia Centro Settentrionale; Pavan et al., 2019). Mean
monthly temperature data are also derived from this dataset. Both mean monthly rainfall depths and mean monthly temperature
data are processed to compute the mean coefficients of the Fourier series that approximate the rainfall and temperature regimes
(4 for rainfall, 4 for temperature, see Appendix A).

This dataset is also used to compute the Mean Annual Precipitation (MAP) and the Mean Annual Temperature (MAT).

Catchment boundaries were used to clip the above-mentioned rainfall and temperature maps and to obtain spatial averages for
the 14 climatological attributes listed in Table 4.

| Climatological attributes | | | |
|---|---|---|---|
| **Attribute** | **Notation** | **Units** | **Description** |
| Mean Annual Precipitation | MAP<br>MAP_cv | mm | Spatial mean and coefficient of variation of the total mean annual precipitation (Braca et al., 2021). |
| B1, B2, C1, C2 | B1_rain, B2_rain,<br>C1_rain, C2_rain | - | Mean values of the coefficients of the Fourier series representative of the rainfall regime computed from the mean monthly precipitation (see Appendix A). |
| Coefficient of variation of rainfall regimes | cv_rain | - | Temporal coefficient of variation calculated from monthly mean rainfall depths derived from Braca et al. (2021). |
| Time step between maximum and minimum of mean monthly rainfall | seas_prec | - | Number of months between the occurrence of the absolute annual maximum rainfall and the subsequent absolute minimum rainfall. |
| Mean Annual Temperature | MAT<br>MAT_cv | °C | Spatial mean and coefficient of variation of the mean annual temperature computed using Braca et al. (2021). |
| B1, B2, C1, C2 | B1_temp, B2_temp,<br>C1_temp, C2_temp | - | Mean values of the coefficients of the Fourier series representation of temperature regimes (see Appendix A). |

**Table 4. List of climatological attributes.**



Once representing the results, one can notice in Figure 6a that the mean annual precipitation (MAP) is higher in mountainous

regions, with largest values in the Alps. The same areas also show low mean annual temperature (MAT; Figure 6b).

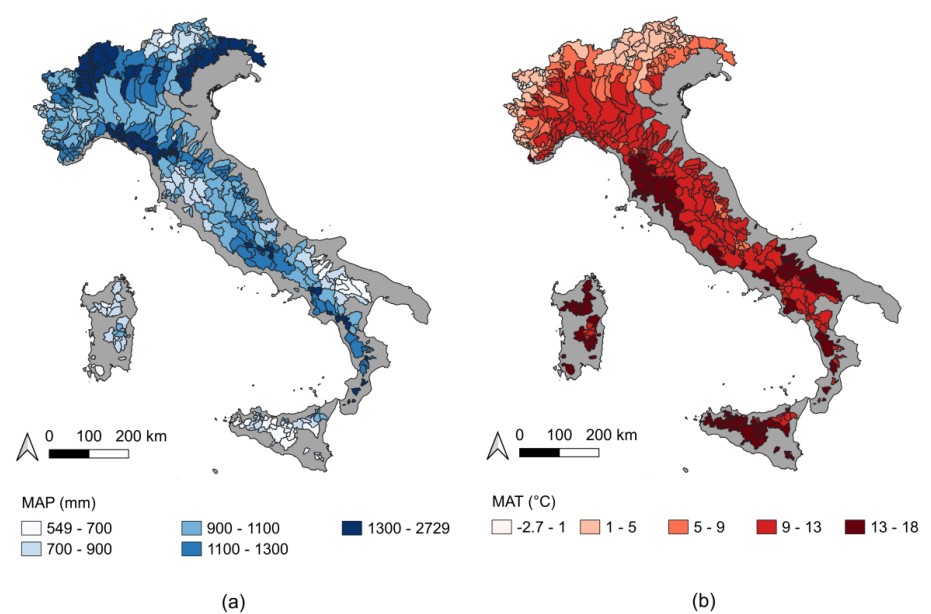

**Figure 6. Spatial representation of the mean annual precipitation (a) and mean annual temperature (b). Administrative boundaries: GADM v3.6.**

**4.3 Extreme rainfall catchment attributes**

With respect to other national databases, like the LamaH-CE dataset (Klingler et al., 2021), one of the main novelties of this

work is the introduction of information related to parameters of the sub-daily precipitation extremes. This inclusion was made

possible thanks to the availability of a rich collection of in-situ data, characterized by a much greater accuracy in capturing

extremes compared to reanalysis data like ERA5 and ERA5-Land datasets (Muñoz Sabater, J., 2019), widely used in the

creation of other datasets. Annual maximum rainfall depths used to derive spatial extreme rainfall statistics are obtained from

the Improved Italian – Rainfall Extreme Dataset (I²-RED; Mazzoglio et al., 2020). The dataset consists of official and quality-

controlled short-duration (1, 3, 6, 12 and 24 hours) annual maximum rainfall depths recorded by more than 5200 rain gauges

over Italy between 1916 and 2019. The stations were subjected to a quality-control procedure to correct errors in the plano-

altimetric positions, duplicates and incorrect rainfall measurements (Mazzoglio et al., 2020). This FOCA dataset represents

the first national-scale collection of mean extreme rainfall catchment attributes. Thanks to FOCA, it is now possible to perform

simple regional and national hydrological studies without the need of retrieving information from 21 different agencies, as

mentioned in the Introduction. Due to the complex data policy that regulates the data collected by the hydrological agencies

(most of them provides the data free of charge only for research purposes, while people interested in using them for commercial

purposes are requested to pay) we were not allowed no include the annual maxima time series in the FOCA dataset. However,

nationwide maps of the parameters allowing the computation of the IDF (Intensity – Duration – Frequency) curves have been

prepared and made freely available within the FOCA dataset.

Rainfall data and related statistics are processed at 250 m resolution with the *autokrige* R function (Hiemstra and Skoien,

2023), that applies an automatic ordinary kriging using the variogram that better fits the data, automatically generated by the

*autofitVariogram* R function. Rainfall statistics obtained with this procedure are:



• the scale factor $a$ and the scaling exponent $n$ of the average Intensity-Duration (ID) curves, obtained by linear regression of the logarithm of the average of rainfall depth $h_d$ over the 1- to 24-hour durations with the logarithm of the duration, where

$$h_d = a \cdot d^n; \qquad (5)$$

• coefficient of L-variation (*L-CV*) of the 1- and 24-hours durations, evaluated with Eq. 6 of Laio et al. (2011);

• coefficient of L-skewness (*L-CA*) of the 1- and 24-hours durations, evaluated with Eq. 7 of Laio et al. (2011).

Time series with at least 10 years of data were used to evaluate the $a$ and $n$ parameters, while series with 20 and 30 years were used for *L-CV* and *L-CA*, respectively. The different record lengths were selected because higher order statistics can not be evaluated from short time series (Koutsoyiannis, 2019). These maps represent the first attempt to reconstruct and represent updated extreme rainfall statistics over the entire Italy, following what has been released in Switzerland (i.e., the Hydrological

Atlas of Switzerland, available on https://hydrologicalatlas.ch/), Austria (i.e., the Hydrological Atlas of Austria, Fürst et al. (2009)), Germany (i.e., the KOSTRA-DWD, or "Coordinated heavy precipitation regionalization and evaluation of the DWD", available on https://www.dwd.de/DE/leistungen/kostra_dwd_rasterwerte/kostra_dwd_rasterwerte.html), the United States (i.e., NOAA Atlas 14, available on https://www.weather.gov/owp/hdsc_currentpf and on https://hdsc.nws.noaa.gov/hdsc/pfds/), the Hawaii (i.e., the Rainfall Atlas of Hawai'i, available on

http://rainfall.geography.hawaii.edu/) and Canada (CSAGroup, 2019).

Table 5 summarizes the characteristics of thr attributes computed and averaged over the 631 catchments.

| Extreme rainfall attributes | | | |
|---|---|---|---|
| **Attribute** | **Notation** | **Units** | **Description** |
| *a* | a<br>a_cv | mm/h | Scale factor of the IDF curve. Mean value and (spatial) coefficient of variation were computed. |
| *n* | n<br>n_cv | - | Scaling exponent of the IDF curve. Mean value and (spatial) coefficient of variation were computed. |
| L-CV *d*h | LCV_*d*h<br>LCV_*d*h_cv | - | Coefficient of L-variation for $d = 1, 3, 6, 12$ and 24-hour duration. Mean value and (spatial) coefficient of variation were computed. |
| L-CA *d*h | LCA_*d*h<br>LCA_*d*h_cv | - | Coefficient of L-skewness for $d = 1, 3, 6, 12$ and 24-hours duration. Mean value and (spatial) coefficient of variation were computed. |

**Table 5. List of rainfall attributes.**

A discussion on the overall spatial and temporal variability of the rainfall extremes is available in Mazzoglio et al. (2020; 2022a; 2022b; 2023). What is useful to comment on here is the result of the spatial averaging of the parameters $a$ and $n$ at the catchment scale (Figure 7a-b). The parameter $a$ can be combined with $n$ to obtain the mean rainfall for different durations (1- to 24-hours). The first one represents the mean 1-hour duration: the higher is $a$, the higher is the intensity of short-duration rainfall extremes over the catchments. This parameter is particularly relevant for small catchments whose characteristic time

lag are of the order of 1 hour. Figure 7 shows a modest correlation between $a$ and elevation (reverse orographic effect; Avanzi et al., 2015; Mazzoglio et al., 2022a; Mazzoglio et al., 2023). One can also notice that the regions where the most severe short-duration rainfall events occur (Figure 7a) are different from those that present higher MAP (Figure 6a), which is a characteristic of the Mediterranean climate (Mazzoglio et al., 2022a).


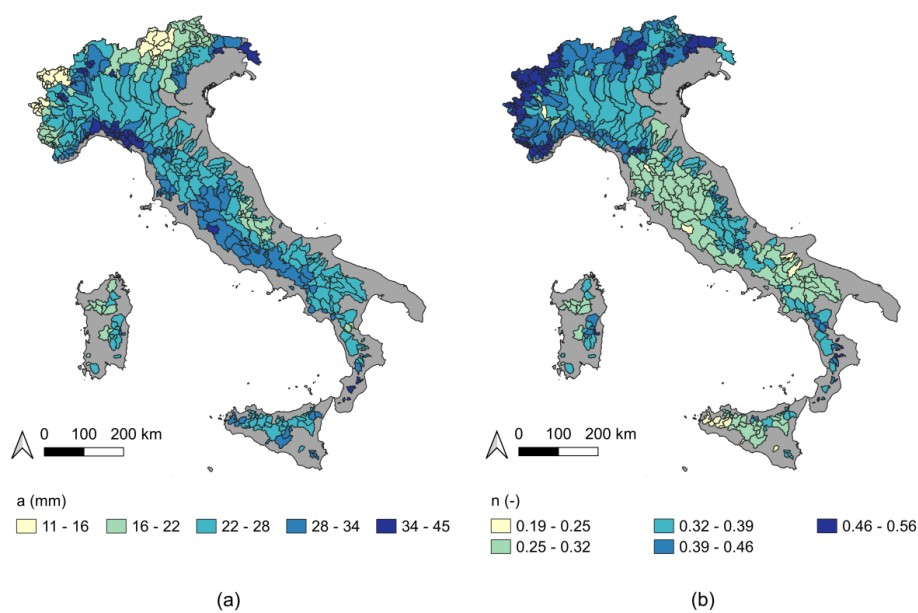

**Figure 7. Spatial representation of the spatial average of the parameter *a* (a) and parameter *n* (b) of the ID curves. Administrative boundaries: GADM v3.6.**

### 4.4 Peak and daily discharge collections

As outlined in Section 2, peak and daily discharges of each gauging station come from Claps et al. (2020a, 2020b, 2020c), which is the most updated systematic flood collection in Italy. Until the 1970s, the data reflect most of the content of the "Pubblicazione n°17" of the SIMN (Servizio Idrografico, 1980). Then, a consistent integration was carried out by merging data from different sources (such as Bencivenga et al., 2011; Barbero et al., 2012; Hall et al., 2015; Brath et al., 2017; Settore Idrologico e Geologico Regionale, 2022; ARPA Lombardia Sistema Informativo Idrologico, 2023), using the same database

set up for the CUBIST project (Claps et al., 2008).

  The provision of this information in this paper (in terms of both the time series and of their statistics, like the mean values of the flood peaks, $Q\_p$) offers the opportunity to fully characterize the climatology of the extremes of the catchments upstream of the gauging stations. The spatial variability of the mean annual flood is shown in Figure 8a, in which one can recognize the presence of nested sub-catchments, resulting from the presence of different measurement stations located along the river path.

This situation is particularly evident in the North-West of Italy, where different small mountainous sub-catchments of the Po basin are highlighted, while the Po basin is the partially visible catchment depicted in black. To have an idea of the length of the discharge time series available one can refer to Figure 8b: the different dot colors show that most of the longer time series (> 50 years) are generally available in North and Central Italy. Moreover, while the reference period covered by the dataset is from 1911 up to 2016, the time series are fragmented and characterized by different time coverage, and thus do not characterize

the entire interval (Figure 9). The decreasing number of time series in recent decades depends on the lack of quality-controlled discharge data published by the regional hydrological agencies, mostly due to the unavailability of updated rating curves. In most cases, the regional agencies are now publishing only water levels, while information about discharges are often missing.


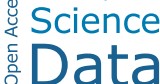

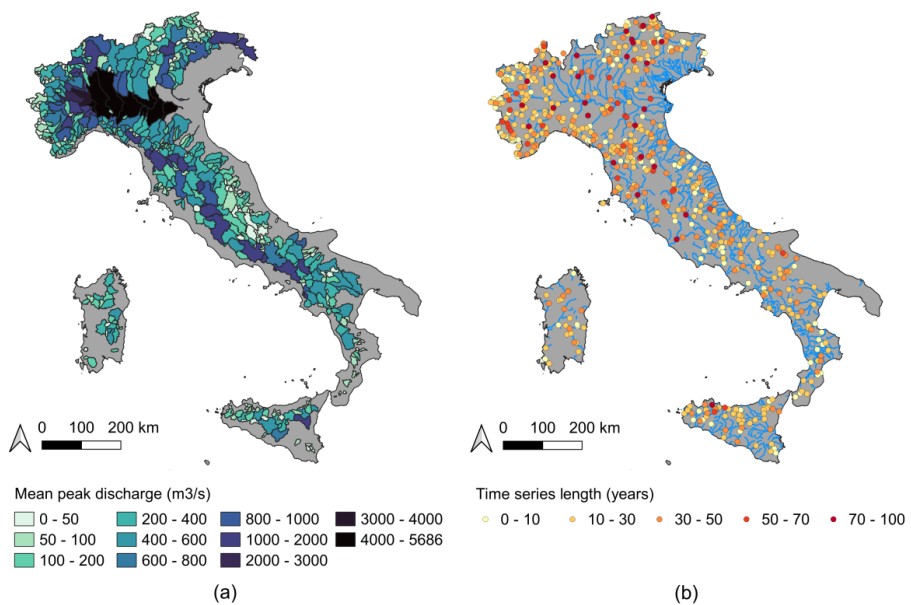

**Figure 8. Spatial distribution of the mean peak discharge (a) and discharge time series length (b). Administrative boundaries: GADM v3.6.**


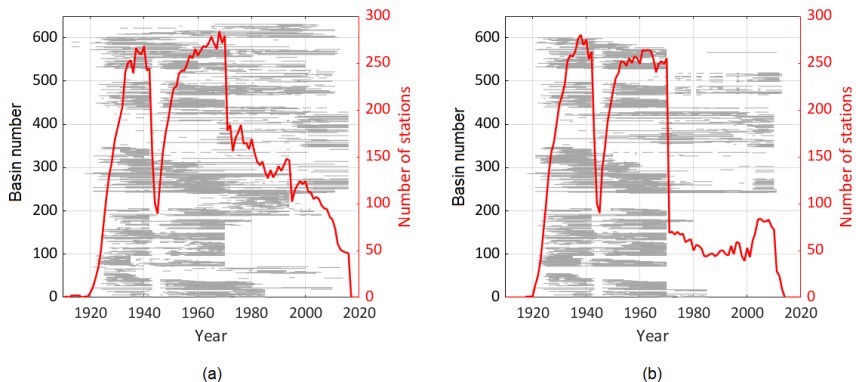

**Figure 9. Temporal availability of the peak (a) and daily (b) discharge. Each grey line represents a time series: gray identify available records, white indicates missing data. The red bold line identifies the total number of yearly measurements available in each year.**

The catchments cover a wide variety of morphological features. Figure 10 shows the empirical cumulative distribution function (ECDF) of the specific mean annual flood (i.e., the ratio between the mean peak discharge $Q\_p$ and the catchment area; Figure 10b). The relation between the area, the specific mean annual flood and the mean catchment elevation is represented in Figure 4, which shows how the Italian catchments are characterized by different hydrological regimes, and a modest dependence of the mean elevation on the specific peak discharge. Catchments with high mean elevation (2000 up to more than 3000 m a.s.l.)

cover a wide range of possible specific mean annual floods, but the interval is smaller than those that characterizes catchments located at lower elevation.

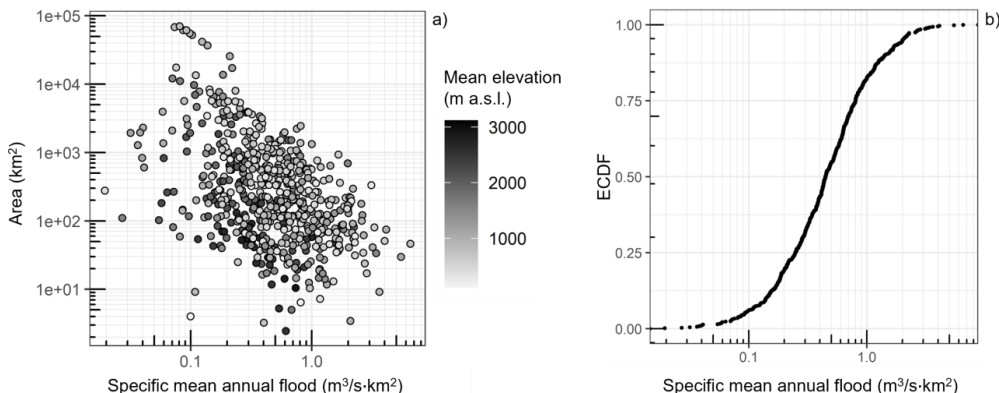


**Figure 10. Relation among area, specific peak discharge and mean elevation of the catchments (a) and empirical cumulative distribution function (ECDF) of the specific mean annual flood (b).**

**5 Conclusions**

In this work we present FOCA (Italian FlOod and Catchment Atlas), a collection of attributes of 631 Italian river catchment

derived from data sources that comply with the following criteria: (a) nationwide coverages; (b) consistency in data quality

(i.e., no regional or local biases); (c) adequacy of the original resolution, in relation to the type of information.

FOCA represents the most updated collection of catchment boundaries and related attributes at the Italian national scale,

summarized in features related to climate (discharge, rainfall and temperature), geomorphology, soil and land use. The dataset

covers an overall area representative of the most landscapes of Italy, including the mountainous ones (elevation ranges from 0

to 4800 m). One of the main novelties with respect to other national-scale datasets is the inclusion of a rich set of

geomorphological and extreme rainfall attributes. The second notable novelty is the inclusion, in the same dataset, of the most

updated information regarding extreme rainfall and discharge.

As mentioned before, Italy has a very wide variability of climate, land and morphological features and this peculiarity emerges

from this new dataset. Figure 2b shows that the FOCA dataset includes catchments with significantly different areas, with

prevailing small-to-medium ones: 105 out of 631 catchments have an area of less than 50 km$^2$ and 300 out of 631 catchments

have an area of less than 200 km$^2$. In addition, 279 out of 631 catchments have a mean elevation higher than 1,000 m a.s.l.,

highlighting that, despite the existence of some large plains, Italy is a mountainous country. On the other hand, on the large

catchments in the North of Italy, the spatial averaging operation over the catchment area becomes less significant, being

evaluated over an area within which significantly different features coexist.

A key decision that we had to took while developing FOCA was whether to use local or global/quasi-global datasets, or both.

The use of global datasets facilitates the comparison of the results obtained in large-scale hydrology works, even when

continental analyses are performed. However, local datasets are, without doubts, characterized by higher-quality information.

In this work we decided to opt for the second approach, selecting for each variable the best possible data, prioritizing local

information, and resorting to global data only in few cases. The availability of the digital information of the catchment

boundaries allows the users to evaluate mean areal values of other variables, as the evapotranspiration and the snow cover,

using their own models or other large-scale reanalysis datasets, like ERA5 and ERA5-Land.

The FOCA dataset provides new opportunities to perform both regional and national-scale studies using catchments shapes

and attributes extracted using a common framework, subjected to the same quality-control procedure. Information about the

algorithms used in this work are also reported in Section 3 to ensure replicability and the calculation of the attributes

characteristics in any ungauged basin. Moreover, the database provides the opportunity to investigate how catchment attributes

control river flows and allows to improve data-intensive investigations as the prediction in ungauged catchments.



## Data availability

The dataset detailed in this paper is available on https://doi.org/10.5281/zenodo.8060737 (Claps et al., 2023). It contains all the catchment boundaries and related catchment attributes described before. To access the latest version of the database, in

case of future updates, the readers can refer to (https://doi.org/10.5281/zenodo.8060736) to download the most recent version.

## Competing interests

The contact author has declared that none of the authors has any competing interests.

## Acknowledgements

This study was carried out within the RETURN Extended Partnership and received funding from the European Union Next-

GenerationEU (National Recovery and Resilience Plan – NRRP, Mission 4, Component 2, Investment 1.3 – D.D. 1243 2/8/2022, PE0000005).

## Author contribution

PC: Conceptualization, Funding acquisition, Methodology, Project administration, Resources, Supervision, Writing – review & editing. GE: Conceptualization, Data curation, Formal analysis, Investigation, Methodology, Software, Validation,

Visualization, Writing – original draft preparation, Writing – review & editing. DG: Conceptualization, Methodology, Writing – review & editing. PM: Conceptualization, Data curation, Formal analysis, Investigation, Methodology, Software, Validation, Visualization, Writing – original draft preparation, Writing – review & editing. IM: Conceptualization, Data curation, Formal analysis, Investigation, Methodology, Software, Validation, Visualization, Writing – original draft preparation, Writing – review & editing.

## Appendix A

The rainfall/temperature/NDVI regimes are described using Fourier series, that allows to reconstruct the shape of the regime using a reduced number of parameters, compared to the 12 different monthly values (or 36 10-day values in the case of the NDVI) used to describe it. According to Fourier's theorem, a wave of period $\tau$ and pulse $\omega = 2\pi/\tau$ can be described as

$$f(t) = A_0 + \sum_{i=1}^{N}(A_i \cos(i\omega t + \phi_i)) \qquad (6)$$

where $t$ is the time, $A_0$ is the mean of $f(t)$ over $t$, $N$ is the number of harmonics, $A_i$ is the harmonic amplitude and $\phi_i$ is the phase. In the case of two harmonics, Eq. 6 can be written as

$$f(t) = A_0 + A_1 \cos\left(\frac{2\pi}{\tau} + \phi_1\right) + A_2 \cos\left(\frac{4\pi}{\tau} + \phi_2\right) = A_0 + A_1 \cos\left(\frac{2\pi}{\tau}t\right) \cdot \cos(\phi_1) - A_1 \sin\left(\frac{2\pi}{\tau}t\right) \cdot \sin(\phi_1) +$$

$$A_2 \cos\left(\frac{4\pi}{\tau}t\right) \cdot \cos(\phi_2) - A_2 \sin\left(\frac{4\pi}{\tau}t\right) \cdot \sin(\phi_2) \qquad (7)$$

By separating the variables that depend on the time from those that are not time-dependent we obtain

$$f(t) = A_0 + B_1 \cdot X_1(t) + C_1 \cdot Y_1(t) + B_2 \cdot X_2(t) + C_2 \cdot Y_2(t) \qquad (8)$$

In this analysis we evaluated $B1$, $B2$, $C1$, $C2$ with the ordinary least squared method (see Eq. 9 to 12).

$$B_1 = A_1 \cos(\phi_1) \qquad (9)$$

$$B_2 = A_2 \cos(\phi_2) \qquad (10)$$

$$C_1 = -A_1 \sin(\phi_1) \qquad (11)$$

$$C_2 = -A_1 \sin(\phi_2) \qquad (12)$$

The equation can be further simplified by assuming as null the mean $A_0$ and solving it for $\tau = 2\pi$.



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
