# Peer review of "FOCA: a new quality-controlled database of floods and catchment descriptors in Italy"

_Earth System Science Data, 2023_

## Referee Comment (RC1)

Review of Manuscript

**'FOCA: a new quality-controlled database of floods and catchment descriptors in Italy'**

By P. Claps et al.

Dear Editor,

I have reviewed the aforementioned work. My conclusions and comments are as follows:

**1.  Scope**

The article is within the scope of ESSD.

**2.  Summary**

In their paper, the authors describe the FOCA data set for Italy, the first nationwide data set of extreme values at 631 river gauges (annual maximum peak discharge and average daily annual maximum discharge) plus attributes of the related basins related to geomorphology, climatology, extreme rainfall, land cover and soil properties.

In the introduction, the authors describe the importance of large-scale, long-term, multi-criteria hydrological data sets and give an overview on recent activities on global, European, and Italian scale. In the same and the following section, the authors explain how the here-introduced data set builds on previous work of (parts of) the authors, and how it goes beyond them, which is mainly in terms of adding to the existing hydrological data climatological, geomorphological, soil and land use attributes. In the following sections, the authors describe in detail the new data set, and how the data were derived and quality-checked, structured by i) geomorphological attributes (section 3), and ii) soil, vegetation, climate, and hydrological data (section 4). In the final section 5, the authors summarize the rationale for compiling the data set, point out how the data set moves beyond previously existing ones, and point to some specialties of the data set.

**3.  Evaluation**

The paper provides a well-structured, self-contained description of the FOCA data set. The authors both show how the data set builds on existing data sets and also how it goes beyond them. Overall, the data set provides real value to the community and the paper deserves publication. The data sources, rationale for data selection, data checks and transformations are clearly described and valid.

A few technical details that require some more attention:

**Data set range**
I think the wide range of catchment sizes and elevation zones covered by FOCA is a valuable and relevant feature, but it is mentioned explicitly only in the conclusion. I recommend mentioning this already in the abstract.

**Catchments near the sea shore**
Fig. 2 (and others) reveal that no catchments with gauges close to the sea shore are included in the data set (grey "rim" along the coast). Why is this so, and does this provide a bias to the data set? Please add an explanation to the text at the appropriate place.

**Uncertainty estimates**

For many applications of the FOCA data set, it will be important to have an (at least rough) estimate of the data uncertainty. For most of the data, the authors provide a clear account of data quality checks, but uncertainty estimates are lacking. This will be especially important for the data derived from observed time series (discharge, precipitation, air temperature), as here related uncertainties are highest. E.g. for the extreme rainfall data, parts of the reporting on uncertainty can be based on the variance of the variogram used for spatial interpolation (lines 362 pp), or for discharge data uncertainty of the water level recordings plus the uncertainty of the rating curve can be used.

**Line 433:** Figure 4 → Figure 10.

**Line 460:** Took → take

**Data set on Zenodo**

I checked the FOCA dataset as published on https://doi.org/10.5281/zenodo.8060737. It contains all the data discussed in the article, the files structure and the formatting of the data is reasonable, understandable and serves the purpose. The file "FOCA_Supporting_Information.pdf" provides a short and concise overview on the data, with one exception: The row raster maps (section 1.5 in the .pdf) just lists the raster maps, but does not provide an explanation of the maps. Please add.

Overall, I think the manuscript is worthy of publication after the above-mentioned **minor revisions** have been done.

Yours sincerely,

Uwe Ehret

---

## Referee Comment (RC2)

The authors have constructed a much-needed dataset on the systematic collection of data on Italian river catchments. This unprecedented dataset at the national scale provides the most complete catchment attributes based on climate, geomorphology, soil, and land use features. This dataset opens opportunities for hydrological studies at regional and national scales such as the control of catchment attributes on the river flows. While the results look convincing and nicely presented, I have several comments that might be helpful for the authors to further improve the manuscript.

**Line 28**, is the name CAMELS an acronym? If so and if it is the first time you used it, please, provide the full meaning of it then you can use only the acronym in the rest of the text. There are several acronyms without the meaning at their first use (e.g., SIMN [line 76], VAPI [line 94], …), please check and provide the meaning.

**Lines 27 to 53**, you started with examples of the dataset at the countries' levels. After that, you mentioned an example at the global scale (line 36), and then you returned with an example of Italy. To be consistent, I will suggest that you start with a global scale example, followed by examples at the countries' levels.

**Line 32**, for all the website links provided in the text (e.g., lines 82, 94, …), please, could you mention the date at which these sites were accessed. This suggestion is valid for all the links in the rest of the text.

**Lines 68 to 73**, this part need to be rewritten. Please, provide a clear objective for each section. Things seem to be merged. In the text, sections 3 and 4 present the same things, even though they are from different sources. To allow readers to better understand the article, I suggest that section 3 presents all the datasets used to derive the different attributes. Section 4 highlights the methodology used to derive those attributes and finally, another section presents all the derived attributes from climate, geomorphology, soil, and land use features.

**Line 120**, I found this text " The database features are presented in two different sections. Section 3 discusses the determination and validation of the attributes depending only on the landforms. Section 4 presents various other features mainly obtained through spatial averaging of rasterized information. " redundant compared to the text at the end of the introduction part.

**Line 124**, section 3.1. What motivates the choice of SRTM and not the other DEM such as Multi-Error-Removed Improved Terrain (MERIT) or Forest And Buildings removed Copernicus DEM (FABDEM) which are considered as digital terrain model (DTM) while SRTM is a digital surface model (DSM) that includes trees and other artifacts? For catchment

delineation or geomorphology study in general, elevation is a crucial parameter. DSM refers to the upper surface of natural and built or artificial features of the environment such as buildings, artificial features, and trees while DTM represents the elevation of the Earth's surface with all natural and built features removed.

**Line 185**, the authors talk about 61 geomorphological attributes in the text, but the table presents almost 36 attributes. How can you explain this?

**Line 189**, the name of the table should be moved above the table not below. Please, check for the tables presented in the text.

**Figure 5**, Fig. 5a stands for the percentage of clc1, not the inverse. Please correct the caption.

**Line 330**, is there a difference between mean monthly rainfall depths and mean monthly precipitation? If there is one, please explain the meaning of each term.

**Line 231**, some typological errors in the sentences need to be corrected. Insert a comma after this sentence" To provide a robust set of catchment features ". Also in other sentences, such as in line 381, correct the word "thr in the". Please, cross-check the entire article.

---

## Author Comment (AC1)

Dear Editor,

I have reviewed the aforementioned work. My conclusions and comments are as follows:

1. Scope

The article is within the scope of ESSD.

2. Summary

In their paper, the authors describe the FOCA data set for Italy, the first nationwide data set of extreme values at 631 river gauges (annual maximum peak discharge and average daily annual maximum discharge) plus attributes of the related basins related to geomorphology, climatology, extreme rainfall, land cover and soil properties. In the introduction, the authors describe the importance of large-scale, long-term, multi-criteria hydrological data sets and give an overview on recent activities on global, European, and Italian scale. In the same and the following section, the authors explain how the here-introduced data set builds on previous work of (parts of) the authors, and how it goes beyond them, which is mainly in terms of adding to the existing hydrological data climatological, geomorphological, soil and land use attributes. In the following sections, the authors describe in detail the new data set, and how the data were derived and quality-checked, structured by i) geomorphological attributes (section 3), and ii) soil, vegetation, climate, and hydrological data (section 4). In the final section 5, the authors summarize the rationale for compiling the data set, point out how the data set moves beyond previously existing ones, and point to some specialties of the data set.

3. Evaluation

The paper provides a well-structured, self-contained description of the FOCA data set. The authors both show how the data set builds on existing data sets and also how it goes beyond them. Overall, the data set provides real value to the community and the paper deserves publication. The data sources, rationale for data selection, data checks and transformations are clearly described and valid.

*R: We thank the reviewer for having appreciated our work.*

A few technical details that require some more attention:

Data set range

I think the wide range of catchment sizes and elevation zones covered by FOCA is a valuable and relevant feature, but it is mentioned explicitly only in the conclusion. I recommend mentioning this already in the abstract.

*R: In the revised version of the manuscript we added this detail in the abstract.*

Catchments near the sea shore Fig. 2 (and others) reveal that no catchments with gauges close to the sea shore are included in the data set (grey "rim" along the coast). Why is this so, and does this provide a bias to the data set? Please add an explanation to the text at the appropriate place.

*R: The reviewer is right: the absence of basins near the coastline is unfortunate and could introduce a bias. However, there is a reason behind this lack, that is systematic in Italy: most of the gauging stations were installed in support to possible hydropower or agricultural or drinking uses of water. The 631 chosen catchments are those for which peak or daily discharges are available, and therefore all included in the "Catalogo delle Piene dei Corsi d'acqua Italiani". But also other catchments, where no flood peaks were available, were positioned relatively far from the coast. We hope that the regional hydrological agencies in charge of the monitoring networks will install hydrometers closer to the coast to be able to further extend the range of basins covered by our dataset in the near future. Some details on this feature are now included in rows 232-236.*

Uncertainty estimates

For many applications of the FOCA data set, it will be important to have an (at least rough) estimate of the data uncertainty. For most of the data, the authors provide a clear account of data quality checks, but uncertainty estimates are lacking. This will be especially important for the data derived from observed time series (discharge, precipitation, air temperature), as here related uncertainties are highest. E.g. for the extreme rainfall data, parts of the reporting on uncertainty can be based on the variance of the variogram used for spatial interpolation (lines 362 pp), or for discharge data uncertainty of the water level recordings plus the uncertainty of the rating curve can be used.

*R: The topic is interesting and assumes different characters if referred to local measurements, such as discharges in river sections, or to areal-averaged values.*

*Starting from local measurements, peak and daily discharges included in the dataset are taken from the official Yearbooks published by the national or regional hydrological agencies. Before publication, these data were validated through application of the corresponding rating curves for every year of measurement. It is not possible to rate the uncertainty of the level-to-discharge conversion without entering very specific technical issues that can be dealt with only in a strict cooperation with the technicians who validated every peak discharge value. This should be done for every single measurement section, as no information about the uncertainty was provided together with the data that we used. This explains why general conclusions cannot be drawn about uncertainty of observed discharge values.*

*We did not include this piece of information in our paper, similarly to what was found in other similar papers related to national datasets. As a matter of fact, information on data uncertainty in the data is, in many cases, very difficult to find.*

*However, we take advantage of this question to better describe the quality of the data used, as specified in the following.*

*Information about monthly rainfall depths, monthly mean temperatures, mean annual precipitation and mean annual temperatures were obtained from a gridded dataset named BIGBANG4.0, produced by the official national reference center of all the regional monitoring agencies in Italy, i.e. the I.S.P.R.A. Institute. The reference network for the BIGBANG4.0 dataset has evolved over time. For mean monthly and annual rainfall depths we were able to find some information on the network density (Figure 6.22 of https://groupware.sinanet.isprambiente.it/bigbang-data/library/bigbang_60/rapporto_ispra_339-21_bigbang_hd/download/en/1/RAPPORTO_ISPRA_339-21_BIGBANG_HD.pdf?action=view). One could notice that the station density is not uniform all over Italy. This means that we can expect local variation in the overall accuracy, with inverse proportion to station density. The interpolation method used was a modified version of the natural neighbours methodology.*

*The dataset of rainfall extremes is characterized by a similar but more uniform data density, especially in some part of North of Italy, as visible in Figure 2 of https://doi.org/10.1080/19475705.2023.2205000. The interpolation method used was an ordinary kriging, as specified in the paper.*

Line 433: Figure 4 → Figure 10.

*R: We thank the reviewer for having highlighted this error. In the revised version of the manuscript we corrected it.*

Line 460: Took → take

*R: Corrected in the revised version.*

Data set on Zenodo I checked the FOCA dataset as published on https://doi.org/10.5281/zenodo.8060737. It contains all the data discussed in the article, the files structure and the formatting of the data is reasonable, understandable and serves the purpose. The file "FOCA_Supporting_Information.pdf" provides a short and concise overview on the data, with one exception: The row raster maps (section 1.5 in the .pdf) just lists the raster maps, but does not provide an explanation of the maps. Please add.

*R: We thank the reviewer for having highlighted this point. We uploaded a revised version of the "FOCA_Supporting_Information.pdf" file with more information on the contents of the raster files.*

Overall, I think the manuscript is worthy of publication after the above-mentioned minor revisions have been done.

Yours sincerely,

Uwe Ehret

---

## Author Comment (AC2)

The authors have constructed a much-needed dataset on the systematic collection of data on Italian river catchments. This unprecedented dataset at the national scale provides the most complete catchment attributes based on climate, geomorphology, soil, and land use features. This dataset opens opportunities for hydrological studies at regional and national scales such as the control of catchment attributes on the river flows. While the results look convincing and nicely presented, I have several comments that might be helpful for the authors to further improve the manuscript.

Line 28, is the name CAMELS an acronym? If so and if it is the first time you used it, please, provide the full meaning of it then you can use only the acronym in the rest of the text. There are several acronyms without the meaning at their first use (e.g., SIMN [line 76], VAPI [line 94], ...), please check and provide the meaning.

R: We thank the reviewer for having highlighted this point. In the revised version of the manuscript we inserted the full meaning of each acronyms (such as CAMELS and VAPI). The meaning of SIMN was already present in row 51.

Lines 27 to 53, you started with examples of the dataset at the countries' levels. After that, you mentioned an example at the global scale (line 36), and then you returned with an example of Italy. To be consistent, I will suggest that you start with a global scale example, followed by examples at the countries' levels.

R: In the revised version of the manuscript we moved the global scale example at the beginning of the section (row 24-27).

Line 32, for all the website links provided in the text (e.g., lines 82, 94, ...), please, could you mention the date at which these sites were accessed. This suggestion is valid for all the links in the rest of the text.

R: In the revised version of the manuscript we inserted the dates.

Lines 68 to 73, this part need to be rewritten. Please, provide a clear objective for each section. Things seem to be merged. In the text, sections 3 and 4 present the same things, even though they are from different sources. To allow readers to better understand the article, I suggest that section 3 presents all the datasets used to derive the different attributes. Section 4 highlights the methodology used to derive those attributes and finally, another section presents all the derived attributes from climate, geomorphology, soil, and land use features.

R: We thank the reviewer for the comment but we suggest that is preferable to keep the current article structure, that is consistent with the one used in most of the articles related to other national datasets, such as the CAMELS ones. In this way, the reader can easily compare all the articles by searching information in the related "common" paragraphs.

Line 120, I found this text "The database features are presented in two different sections. Section 3 discusses the determination and validation of the attributes depending only on the landforms. Section 4 presents various other features mainly obtained through spatial averaging of rasterized information. " redundant compared to the text at the end of the introduction part.

R: We agree with the reviewer: in the revised version of the manuscript we removed this redundant part.

Line 124, section 3.1. What motivates the choice of SRTM and not the other DEM such as Multi-Error-Removed Improved Terrain (MERIT) or Forest And Buildings removed Copernicus DEM (FABDEM) which are considered as digital terrain model (DTM) while SRTM is a digital surface model (DSM) that includes trees and other artifacts? For catchment delineation or geomorphology study in general, elevation is a crucial parameter. DSM refers to the upper surface of natural and built or artificial features of the environment such as buildings, artificial features, and trees while DTM represents the elevation of the Earth's surface with all natural and built features removed.

R: When considering the choice of digital elevation/terrain model for catchment delineation or geomorphology studies, several factors come into play. While it's true that elevation is a crucial parameter for such studies, the choice between different DEMs/DTMs depends on the specific requirements and objectives of the research. In this work we decided to use SRTM to be consistent with previous research works performed over Italy, even if we are aware of the limitation of this dataset. Moreover, FABDEM was not available at the time of the realization of this dataset.

Line 185, the authors talk about 61 geomorphological attributes in the text, but the table presents almost 36 attributes. How can you explain this?

R: The total number of attributes that we computed is 61 even if, for the sake of brevity, they are collapsed into 36 different names. As an example, the elevation values of the hypsographic curve (i.e., the curve that defines the distribution of catchment areas located above a specific elevation) are summarized in the attribute "elev\_x" with x = 2.5, 5, 10, 25, 50, 75, 90, 95 and 97.5% of the basin area. This means that for "elev\_x" we have 9 different descriptors, one for each percentage. The same applies also for other descriptors, such as "HS\_num\_u" and "width\_mean". So, in the end, the total number is 61.

Line 189, the name of the table should be moved above the table not below. Please, check for the tables presented in the text.

R: In the revised version of the manuscript we moved all the descriptions above the tables.

Figure 5, Fig. 5a stands for the percentage of clc1, not the inverse. Please correct the caption.

R: We thank the reviewer for having pointed out this error. In the revised version of the manuscript we corrected the caption.

Line 330, is there a difference between mean monthly rainfall depths and mean monthly precipitation? If there is one, please explain the meaning of each term.

R: In that section we used the word "rainfall" as synonym of "precipitation" to avoid repetition. In order to avoid misunderstandings, in the revised version of Section 4.2 we refer to precipitation.

Line 231, some typological errors in the sentences need to be corrected. Insert a comma after this sentence" To provide a robust set of catchment features ". Also in other sentences, such as in line 381, correct the word "thr in the". Please, cross-check the entire article.

R: We thank the reviewer for having highlighted these errors: we have carefully checked the English and in the revised version of the manuscript we corrected them. We have also asked an English native speaker to revise the manuscript.

---

## Referee Report (RR1)

Dear Editor,

Thank you to nominate me to review once again the manuscript essd-2023-233.

The authors have responded correctly to all my comments, which have improved their manuscript. I therefore support the publication of this manuscript.

Sincerely,

---

## Author Response (AR2)

Dear Editor,

we reviewed the article on the basis of the comments that we received.

We added a paragraph at the beginning of Section 4 about data uncertainty (rows 264-281). In this section we divided the datasets that were computed from other research groups from those that were produced by the authors. When revising this section, we moved at the beginning of Section 4 two sentences that were previous inserted in Section 4.1.

For rainfall extremes, we followed the suggestion of the reviewer and we added a description of the kriging standard deviation and one figure at the end of Section 4.3 (rows 420-427). We preferred the standard deviation of the kriging estimates in place of the variance of the kriging estimates to facilitate the comparison with the values of Figure 7a.

Also for discharge extremes we added some sentences at the beginning of Section 4.4 (rows 437-442).

Best regards,

Paola Mazzoglio on behalf of all authors

---

## Author Response (AR3)

Dear Editor,

regarding the Supporting Information, in Section 1.1 we mentioned that the discharge values are expressed in m3/s. In Section 1.4 we inserted an extended version of the tables already reported in the manuscript. These tables were enriched by a new field that contains, for each catchment attribute, the corresponding column number of the Excel file.

We thank you for this suggestion, now the Supporting Information should be more understandable!

We also addressed the comment received from the Editorial Support about the inclusion of a reference of a preprint (Höge et al.). We modified it since the paper has now been accepted for publication in ESSD.

Best regards,

Paola Mazzoglio on behalf of all authors